# Presenteeism, Psychosocial Working Conditions and Work Ability among Care Workers—A Cross-Sectional Swedish Population-Based Study

**DOI:** 10.3390/ijerph17072419

**Published:** 2020-04-02

**Authors:** Klas Gustafsson, Staffan Marklund, Constanze Leineweber, Gunnar Bergström, Emmanuel Aboagye, Magnus Helgesson

**Affiliations:** 1Department of Clinical Neuroscience, Division of Insurance Medicine, Karolinska Institutet, SE-171 77 Stockholm, Sweden; staffan.marklund@ki.se (S.M.); magnus.helgesson@ki.se (M.H.); 2Stress Research Institute, Stockholm University, SE-171 77 Stockholm, Sweden; constanze.leineweber@su.se; 3Department of Occupational Health Sciences and Psychology, Faculty of Health and Occupational Studies, Centre for Musculoskeletal Research, University of Gävle, SE-801 76 Gävle, Sweden; gunnar.bergstrom@hig.se (G.B.); emmanuel.aboagye@hig.se (E.A.); 4Institute of Environmental Medicine, Unit of Intervention and Implementation Research for Worker Health, Karolinska Institutet, SE-171 77 Stockholm, Sweden

**Keywords:** presenteeism, psychosocial working conditions, nurses, care workers, health, work ability

## Abstract

Presenteeism, attending work while ill, has been examined in different contexts in the last few decades. The aim was to examine whether poor psychosocial working conditions and perceived work ability are associated with increased odds ratios for presenteeism, focusing on nursing professionals and care assistants. A cross-sectional population-based study was conducted. The selected individuals were extracted from representative samples of employees, aged 16–64, who participated in the Swedish Work Environment Surveys between 2001 and 2013 (*n* = 45,098). Three dimensions of psychosocial working conditions were measured: job demands, job control, and job support. Presenteeism and perceived work ability was measured. Using multiple logistic regression analyses, odds ratios for presenteeism with 95% confidence intervals (CI) were estimated. While nurses (*n* = 1716) showed the same presenteeism level as all the other occupation groups (*n* = 37,125), it was more common among care assistants (*n* = 6257). The odds ratio for presenteeism among those with high job demands (OR = 2.37, 95% CI 2.21–2.53), were higher among women than among men. For nursing professionals and care assistants, the odds ratios for presenteeism were highest among those with the lowest work ability level. The problems of presenteeism and low work ability among many health and care workers may be lessened by a reduction in psychosocial demands.

## 1. Introduction

Sickness presenteeism, attending work while ill, has increased during the last few decades [1,2], especially among nurses and care assistants [3,4,5] and other healthcare employees [6,7,8].

A recent systematic review showed that a wide variety of explanatory factors contribute to the prevalence of presenteeism, including psychosocial working conditions, employment conditions, and factors related to sickness insurance [9]. According to a Dutch study, there is an indication that job demands and burnout exhibited a substantial longitudinal relationship with presenteeism among nurses [10].

When it comes to sex-related differences in presenteeism, previous studies have provided inconsistent results. Some studies indicated that women report a higher prevalence of presenteeism than men [2,3,11,12], while other studies showed no sex differences [5,13].

Previous research has shown that presenteeism may affect future health negatively [9,10]. A higher incidence of presenteeism among nurses may therefore lead to more negative health consequences, which in the long run could result in negative consequences for health organisations and patients [14]. Working while ill may result in low work ability, inefficient work, and an increase in errors, which may lead to productivity loss and, in the case of healthcare personnel, to lower medical safety for patients [15,16,17]. The costs of presenteeism have been examined in different healthcare contexts, with the findings indicating that presenteeism can adversely affect the quality of patient care and negatively affect patient safety [18], worsen the spread of infectious diseases [19], and delay patient healing [1]. Moreover, Finnish studies have found presenteeism to reduce nurses’ work ability [20] and showed that perceived work ability among employees in a food factory may be a robust indicator for assessing perceived productivity loss [21].

The objective of the present study was to examine the extent to which psychosocial working conditions are associated with increased odds ratio of presenteeism, focusing on two occupational groups, nursing professionals and care assistants, and sex-related differences. A specific aim was to examine how perceived work ability was related to presenteeism.

## 2. Methods

### 2.1. Ethic Aspects

The research protocol was approved by the Regional Research Ethics Committee of Stockholm, Sweden in 2015 and 2018 (No: 2015/2203-31/5, No. 2018/223-31/5 and No: 2018/5:2). Other researchers may obtain the same data in the same manner as we did from Statistics Sweden, URL: http://www.scb.se/.

### 2.2. Study Design and Participants

A cross-sectional study based on seven surveys of the working population in Sweden between 2001 and 2013 was conducted. Data were obtained from the Swedish Work Environment Surveys (SWES), conducted every second year since 1989 by Statistics Sweden (SCB) on behalf of the Swedish Work Environment Authority [22]. SWES participants were sampled from the Labor Force Surveys (LFS) conducted by Statistics Sweden. Persons who were asked to respond to the LFSwere randomly selected from the entire Swedish population, stratified by county, sex, citizenship, and employment status. Randomly selected respondents to the LFS, who were gainfully employed and between 16 and 64 years old, were first contacted by telephone; those willing to participate then received a self-completion questionnaire and these respondents constituted the SWES subsample. The current study was based on data from 45,098 working men and women who participated in any biannual wave of the SWES between 2001 and 2013. The participants from the SWES were added to the cohort, and the period for each sub cohort started the year after the interview on January 1st (2001–2013). The annual response rates varied between 77% and 66% [22].

### 2.3. Measurements

#### 2.3.1. Classification of Occupations

The occupations included in the present study, classified according to the 1996 Swedish Standard Classification of Occupations (SSYK96), were organized into two groups: (1) nursing and midwifery professionals and nursing associate professionals (SSYK 223 and 323; *n* = 1716), and (2) personal care and related workers (SSYK 513; *n* = 6257) (www.scb.se, Swedish Standard Classification of Occupations). The first occupational group, “nursing professionals,” consists of specialized and non-specialized registered nurses with a university degree working in hospitals and other healthcare organizations. The second, labeled “care assistants,” consists of personal care and related workers, including assistant nurses, hospital ward assistants, and home-based personal care workers and assistants in childcare. The required educational level for these occupations is generally secondary school level. All of the other occupations in the SWES (*n* = 37,125) were also included and served as a comparison group.

#### 2.3.2. Presenteeism

Data on presenteeism were obtained from the SWES, 2001–2013 [22]. The following item was chosen to indicate presenteeism [3,5,10]: “How many times during the past 12 months have you worked, even though you really should have not worked given your medical condition?” The items were answered on a four-point response scale: never (*n* = 13,395; 30%), once (*n* = 9362; 21 %), two or three times (*n* = 14,454; 32%), and four times or more (*n* = 7600; 17%) as well as missing responses (*n* = 287; 1%). The responses were dichotomized for analysis into ‘no presenteeism’ (‘never’ or ‘once’) (*n* = 22,757; 51%) and ‘presenteeism (‘two or three times’ or ‘four times or more’) (*n* = 22,054; 49%).

#### 2.3.3. Psychosocial Working Conditions

Data on psychosocial working conditions were obtained from the SWES [22]. Aspects of working conditions, which were included in the study, covered job demands, job control, and job support. 

Job demands and job control were measured by a number of items which served as proxy indicators applicable to the demand-control model [23] as formulated by Karasek and collaborators [24]. Job support from supervisors and from colleagues was each measured by one item. More detailed information on the psychosocial items, the response alternatives, and the coding can be found in a previous publication by the research team [25].

Job demands were captured by the following four items: ‘Is your work so stressful that you do not have time to talk or even think about something other than work?’‘Does your work require your full attention and concentration?’‘Do you have so much work that you have to miss lunch, work late or take work home?’‘I have far too much to do at work?’

Job control was measured by four items capturing aspects of work pace and work content: ‘Do you have the opportunity to determine your work pace?’‘Are you able to determine when various working duties are to be carried out (for example, by choosing to work a bit faster on some days and taking it easier on other days)?’‘Do you participate in decisions on the arrangement of your work (e.g., what is to be done, how to do it or who will work with you)?’‘I have too little influence at work?’

Two items captured job support from supervisors and colleagues: ‘Are you able to get support and encouragement from supervisors, when work feels difficult?’‘Are you able to get support and encouragement from colleagues, when work feels difficult?’

All response scales were dichotomized closest to the upper quartile to indicate the most adverse conditions. For the item concerning whether their work required attention and concentration, almost half of the responses were in the ‘no’ coded category, and it was therefore not possible to use the upper quartile as the cut-off point. Those who answered ‘no’ on the items comprised the reference group for the analyses. 

#### 2.3.4. Work Ability

The following item was chosen as an indicator of work ability [22]: 

“We assume that your ability to work, when it was best, is valued at 10 points. How many points would you give your current ability to work? (Please check the appropriate number (1–10 points), 10 points means that your ability to work is at its best right now).” Responses were originally given on a ten-point scale and later coded into five categories, where 1–6 points was seen to indicate low work ability, 7, 8, and 9 points were reported as the original scale, and 10 points was seen to indicate high work ability. The reason for collapsing categories 1–6 into one category was that only 5.2% of the study population reported these low values. Since this item is only available from 2007 onwards, a reduced sample was used for the analyses including work ability (subsample, *n* = 24,902).

### 2.4. Potential Confounders

#### 2.4.1. Sociodemographic Factors

Sex, age at interview (16–29, 30–39, 40–49, and 50–64 years), education (≤9 years, 10–12 years, >12 years), country of birth (born in Sweden or other country), and employment sector (public sector vs. private sector) were selected as potential confounders. The data were all obtained from The Longitudinal Database for Health Insurance and Labor Market Studies (LISA). 

In addition to these factors, two indicators of negative physical work exposure, three health indicators, and two indicators of long-term sickness absence and disability pension were also used as confounders.

#### 2.4.2. Physical Working Conditions

The following two items were chosen as indicators of heavy physical work and strenuous work postures. More detailed information on the items capturing physical work and on the response alternatives and coding can be found in previous publications by the research team [26].

Heavy physical work—‘Are you required to lift at least 15 kg at a time several times per day?’Strenuous work postures—‘Do you bend or twist yourself in your work in the same way repeatedly for an hour, for several hours during the same day?’

The responses to the five-point response scale (every day, 1 day of 2, 1 day of 5, 1 day of 10, not at all) were dichotomized closest to the upper quartile to indicate the most adverse conditions. The responses were dichotomized into ‘No’ (≤1 day of 5) and ‘Yes’ (≥ 1 day of 2), and ‘Yes’ (every day).

#### 2.4.3. Self-Rated symptoms

Three items were chosen as indicators of health symptoms:“Have you experienced pain in your upper back or neck after working, during the past three months?”“Have you had trouble sleeping during the last three months?”“Have you felt tired and listless during the last three months?”

The responses to the five-point response scale on self-rated symptoms (every day, 1 day of 2, 1 day of 5, 1 day of 10, not at all) were dichotomized closest to the upper quartile to indicate the most adverse conditions. The responses were dichotomized into ‘No’ (≤1 day of 5) and ‘Yes’ (≥1 day of 2), and ‘Yes’ (every day). The ‘No’ group was then used as the reference category in the analyses.

#### 2.4.4. Sickness Absence and Disability Pension

Long-term sickness absence (LTSA) during the calendar year in which the individual was included in the SWES (spanning 2001–2013) was operationalized as the number of spells of the individuals medically certified sickness absence lasting 60 days or more, as recorded in the Swedish Social Insurance registers and obtained from the LISA database. The categories are ‘No’ (<60 days) and ‘Yes’ (≥60 days). 

All cases of granted full-time or part-time disability pension during the years following participation in the surveys (2002–2014) were included, regardless of the diagnostic category.

### 2.5. Statistical Analyses

The selected participants from the SWES surveys were consecutively added to the cohort (December 31, 2001–2013). The odds ratios (ORs) for presenteeism, with 95% confidence intervals (CIs), were estimated using multiple logistic regression analysis. The statistical analyses were conducted in two steps. First, the ORs of the psychosocial variables and confounders of presenteeism were analyzed, one by one, adjusting for age (one-year intervals) and year of interview. Second, the associations between presenteeism and both psychosocial exposures and work ability were analyzed and stratified by sex and occupation into the categories of nursing professionals, care assistants, and all other occupations. Adjustment for two sets of confounders were made: Model 1, age at interview and year of interview and Model 2, age at interview and year of interview plus sociodemographic conditions, sector of employment, physical work conditions, health symptoms, sickness absence, and disability pension. All statistical analyses were conducted with SAS, version 9.4., statistical software (SAS Institute, Inc., Cary, North Carolina).

## 3. Results

Table 1 shows that care assistants went to work while ill significantly more often than nursing professionals and those in the all other occupations category according to the self-report analysis (OR = 1.20, 95% CI 1.14–1.23). Furthermore, presenteeism was significantly related to being a woman, middle aged, less educated, employed in the public sector, and foreign born.

Table 1 also indicates a dose–response relationship between presenteeism and perceived work ability.

Table 2 contains information and findings about psychosocial working conditions and health indicators that may be related to presenteeism among the total study group. It shows highly significant ORs for three indicators of high psychosocial job demands; that is, “stressful—no time to think,” “much work—miss lunch,” and “far too much to do.” Significant ORs were also noted for all indicators of low job control such as “determine work pace” and “influence.” Similarly, little or no support from supervisors and from colleagues was associated with presenteeism. Furthermore, strenuous work postures and heavy lifting were also associated with presenteeism, as were negative health symptoms, LTSA (≥60 days), and being granted a disability pension.

### 3.1. Analyses Stratified by Sex

As shown in Table 3, female nursing professionals, when compared to females in the all other occupations group, showed significantly lower adjusted ORs for presenteeism (OR = 0.82, 95% CI 0.73–0.91). Such comparison among the males revealed no significant differences between the occupational groups. The adjusted ORs for presenteeism in the total study group were higher among the women than the men with high work demands, such as “far too much to do” (OR = 2.37). However, a comparison of the associations found between presenteeism and each of the four job control categories within each sex shows similar figures among the categories. This was also the case for the associations between presenteeism and poor job support. Table 3 also shows that presenteeism was strongly associated with low work ability among both women and men in the total study group, even after adjustment for confounders. The ORs for presenteeism increased gradually as the work ability points decrease from nine to seven points. The OR for presenteeism was 1.87 among women with low work ability (one to six points), and 1.58 among men with low work ability (see Table 3, model 2).

### 3.2. Analyses Stratified by Occupation

Table 4 shows analyses stratified by the three occupational groups. All three occupational groups showed significant associations between high psychosocial demands and presenteeism, even after controlling for confounders. Two indicators of low job control, inability to determine work pace and lack of influence, were associated with presenteeism. Low influence was highly associated with presenteeism among nursing professionals (OR = 1.61, 95% CI 1.28–2.02) and care assistants (OR = 1.51, 95% CI 1.34–1.71). Also, a lack of support from supervisors, but not from colleagues, had a significant association with presenteeism among both nurses and care assistants.

In the all other occupation group, both a lack of support from supervisors and from colleagues increased the OR estimates of presenteeism significantly, but among nursing professionals and care assistants the increase in support from colleagues was not significant when adjustment for confounders was introduced. In all three groups, presenteeism was significantly related to low work ability, with particularly high ORs of presenteeism among nursing professionals reporting low work ability (OR = 3.82, 95% CI 1.57–10.77). Among care assistants, the associations between presenteeism and work ability were less pronounced, albeit significant, and those reporting the lowest level of work ability had an OR of 1.97 (95% CI 1.40–1.79). There were no significant differences between women and men for any of the three occupational groups when the selected confounders were included in the analyses.

## 4. Discussion

The present study examined the relationships between presenteeism and both psychosocial working conditions and perceived work ability. The focus was on sex differences and the two occupational groups of nursing professionals and care assistants.

The study showed that presenteeism was most common among care assistants and that its prevalence among nursing professionals did not deviate from that among the all other occupations group. The high prevalence of presenteeism found among care assistants is in line with findings from studies from other countries [3,4,5,6,11,20,27,28,29]. Unlike the present study, other studies have found that nursing professionals have higher levels of presenteeism than other occupations [7,9,16,17,30].

An interesting sex-related finding was that the higher levels of presenteeism found among women in the univariate analyses were no longer found for any of the three groups after controlling for confounders in the stratified analyses. Previous studies have reached contradictory conclusions concerning the differences between women and men, with some showing no differences in the prevalence of presenteeism between the sexes [5,13] and others reporting higher prevalence among women [2,3,11,12]. The reason why the gender differences in presenteeism rates disappeared after confounder control is that factors such as education, exposure to heavy physical work and health symptoms are included among the confounders and that there are sex differences in the prevalence of these factors.

The fact that younger nurses and care assistants had lower levels of presenteeism than their older counterparts may be related to the fact that their employment security is weaker, but it may also be related to their lower income security or lower level of loyalty towards patients or employees.

Our results showing associations between presenteeism and psychosocial working conditions, particularly high job demands and low job support, are supported by a large number of previous studies of healthcare and social care employees [2,4,7,9,10,11,12,13,20]. It should be noted, however, that the prevalence of presenteeism has been rather stable over time among health and care employees as well as among the general working population (Statistics Sweden, Swedish Work Environment Survey, scb.se) [22], despite a marked increase in demanding psychosocial conditions in the last few decades in Sweden [26]. This can indicate that there may also be compensating factors at work or other factors that affect the individuals’ choice of presenteeism that have not been included in the present study. Increased employment insecurity may, for example, increase presenteeism, while increased sickness absence may mean lower presenteeism. Also, the finding of presenteeism being related to previous, present, and future health problems has also been reported in a number of studies [2,3,9,11,12,14,20,31,32,33]. A couple of studies have shown that intention to leave one’s occupation and risk of disability pension are higher among employees in health and care occupations where presenteeism is prevalent [25,34].

The study found strong associations between presenteeism and low work ability and it was remarkable that the risk of presenteeism among those who reported the lowest degree of work ability was almost fourfold higher among nurses and twofold higher among care assistants. These associations remained even after controlling for ill-health symptoms. Associations between presenteeism and reduced work ability is in line with studies from other countries, although different definitions of work ability and productivity have been applied in the studies [1,7,8,13,15,16,21,30,35,36]. One study that focused on the relationship between reduced work ability and productivity loss argued that perceived work ability is a robust indicator for assessing productivity loss [21]. Thus, the relatively high level of presenteeism in healthcare occupations and the fact that a large number of employees who engage in presenteeism report very low work ability indicates that many employees in this sector are not working to the best of their abilities. Additionally, presenteeism has been shown to increase the risk of spreading infectious diseases [19], a potentially detrimental hazard in health and care settings [17,18].

### Strengths and Limitations

In the current study, the number of interviews was large and based on representative samples from the working population, and the response rates to the questionnaires were satisfactory. Since poor health is the root of sickness presenteeism, individuals’ health symptoms, sickness absence and disability pension status were controlled for. To the best of our knowledge, no other Swedish studies have investigated presenteeism in relation to work ability among care workers. The present study has some limitations. One is that it was not possible to draw causal conclusions because of the cross-sectional design in which working conditions and self-reported health symptoms were assessed at the same point in time as presenteeism. Still, presenteeism was measured through a retrospective question referring to the previous year, while the other factors concerned the present situation. Although the associations between reduced health and presenteeism and between reduced health and low work ability may seem trivial, the causal mechanisms are not well known. One reason for this is the lack of information on how health conditions are assessed by the individuals when they decide to go to work ill [1,5,12].

## 5. Conclusions

The study indicated that presenteeism among healthcare and care employees, as well as among other occupations, was associated with high job demands and lack of supervisor job support. There were also extremely strong associations between presenteeism and low work ability. This may affect productivity and safety in healthcare organizations as well as the present and future health (and sickness absence) of its employees. It may also be a health risk for the patients. In order to reduce presenteeism and low work ability, and thus the problems they lead to, efforts should be made to improve psychosocial working environments. Reduction in job demands and improvements in supervisory support, as well as educational and psychological measures to give nurses and care assistants better strategies to meet stress at work, are essential both for the individuals and for the health and care organizations.

## Figures and Tables

**Table 1 ijerph-17-02419-t001:** Number of individuals, prevalence, and odds ratios (ORs) of presenteeism (two times or more) with 95% confidence intervals (CIs), related to sociodemographic variables (sex, age, education)*,* employment sector, country of birth, occupation, and work ability, interviewed 2001–2013 ^a^.

				Presenteeism ^a^			
		Men	Women	Univariate			
	*P* ^b^	*N* ^c^	*N* ^c^	*P* ^d^	OR ^e^	CI	
Occupation							
All other occupations	82.3	19,271	17,854	48.6	1		
Nursing professionals	3.8	143	1573	47.5	0.97	0.88	1.07
Care assistants	13.9	616	5641	53.5	**1.20**	1.14	1.23
Sex							
Men		20,030	-	47.5	1		
Women		-	25,068	50.6	**1.13**	1.08	1.17
Age at interview (years)							
16–29	12.9	2512	3288	52.0	1		
30–39	22.0	4571	5380	53.3	**1.27**	1.17	1.39
40–49	26.7	5251	6776	51.6	**1.44**	1.28	1.63
50–64	38.4	7696	9624	44.3	**1.35**	1.13	1.62
Level of education							
University or college	42.4	7529	11,589	48.0	1		
High school	47.3	9885	11,445	50.2	**1.08**	1.04	1.13
Compulsory	10.3	2602	2023	49.9	**1.16**	1.09	1.24
Employment sector							
Private	53.2	14,268	9507	48.1	1		
Public	46.8	5592	15,354	50.6	**1.14**	1.10	1.18
Country of birth							
Sweden	92.1	18,548	22,980	48.6	**1**		
Other country	7.9	1479	2088	56.4	**1.38**	1.29	1.48
Work ability ^f^							
High, 10 points	52.8	5979	7167	43.6	1		
9 points	11.9	1516	1456	46.1	**1.14**	1.05	1.24
8 points	21.2	2578	2696	51.1	**1.44**	1.35	1.54
7 points	8.6	1055	1088	57.9	**1.92**	1.75	2.11
6 points	2.2	235	313	67.5	**2.92**	2.43	3.51
5 points	2.2	192	344	63.0	**2.48**	2.07	2.97
4 points	0.5	43	71	71.9	**3.65**	2.44	5.58
3 points	0.3	36	47	69.5	**3.32**	2.09	5.42
2 points	0.1	13	23	77.8	**4.78**	2.28	11.27
Low, 1 point	0.2	22	28	75.0	**3.94**	2.11	7.93

^a^ The study group (*n* = 45,098). ^b^ Prevalence (P) of individuals (%). ^c^ Number of individuals (*n*). ^d^ Prevalence (P) of presenteeism, two times or more (%). ^e^ Odds ratio (OR) of presenteeism with 95% confidence interval (CI), adjusted for age (continuous variable) and year of interview. ^**f**^ Sub-sample 2007–2013 *n* = 24,902. Bold = statistically significant at the *p* < 0.05 level.

**Table 2 ijerph-17-02419-t002:** Number of individuals, prevalence, and odds ratios (ORs) of presenteeism (two times or more) with 95% confidence intervals (CIs), related to psychosocial and physical work factors, health symptoms, long-term sickness absence, and disability pension, interviewed 2001–2013 ^a^.

	Presenteeism ^a^					
	Men	Women	Univariate			
	*N* ^b^	*N* ^b^	*P* ^c^	OR ^d^	CI	
Job demands						
Stressful—no time to think, Yes (≥3/4 of working time)	3402	6604	65	**2.32**	2.21	2.43
Attention/concentration, Yes (≥3/4 of working time)	7936	12,483	53	**1.39**	1.34	1.45
Much work—miss lunch etc., Yes (≥1 day of 2)	4161	4770	64	**2.16**	2.06	2.27
Far too much to do, Yes (agree, partly agree)	3019	5058	67	**2.48**	2.36	2.62
Job control						
Determine work pace, No (≤1/10 of working time)	3390	7400	58	**1.59**	1.52	1.66
Determine working duties, No (no, not at all)	2306	5355	53	**1.22**	1.16	1.28
Participate in decisions, No (≤mostly not)	4536	6821	52	**1.16**	1.11	1.21
Influence, No (disagree, partly disagree)	4811	6555	58	**1.58**	1.52	1.65
Job support						
Support from supervisors, Mostly not—never	7472	7753	58	**1.75**	1.68	1.82
Support from colleagues, Mostly not—never	3964	2914	57	**1.54**	1.46	1.62
Physical work						
Strenuous work postures, Yes (every day)	3889	5992	59	**1.63**	1.56	1.71
Heavy lifting, Yes (≥1 day out of 2)	5312	4232	60	**1.73**	1.65	1.81
Health symptoms						
Upper back or neck pain, ≥1 day of 2	3441	7383	67	**2.67**	2.55	2.79
Tired and listless, ≥1 day of 2	3187	6082	72	**3.33**	3.17	3.51
Sleeping troubles, ≥1 day of 5	3705	5871	68	**2.86**	2.73	3.00
Long-term sickness absence, Yes (≥60 days)	492	1268	67	**2.31**	2.09	2.56
Disability pension after interview, Yes	287	705	67	**2.33**	2.04	2.68

^a^ The study group (*n* = 45,098). ^b^ Number of individuals (*n*). ^c^ Prevalence (P) of presenteeism, two times or more (%). ^d^ Odds ratio (OR) of presenteeism with 95% confidence interval (CI), adjusted for age (continuous variable) and year of interview. Bold = statistically significant at the *p* < 0.05 level.

**Table 3 ijerph-17-02419-t003:** Associations between presenteeism (two times or more) and occupation, work ability, and psychosocial variables stratified by sex. Number of cases and odds ratios (ORs) with 95% confidence intervals (CIs) are presented.

	Presenteeism ^a^									
		Men (*n* = 20,030)					Women (*n* = 25,068)			
	*n* ^b^	OR ^c^	OR ^d^	CI	CI	*n* ^b^	OR ^c^	OR ^d^	CI	CI
Occupation										
All other occupations	10085	1	1			8889	1	1		
Nursing professionals	72	1.05	1.07	0.75	1.54	823	0.92	**0.82**	0.73	0.91
Care assistants	297	1.13	1.08	0.90	1.29	2591	**1.16**	0.97	0.90	1.04
Job demands										
Stressful—no time to think, Yes (≥3/4 of working time	1198	**2.39**	**1.71**	1.57	1.87	2333	**2.25**	**2.11**	1.98	2.24
Attention/concentration, Yes (≥3/4 of time)	3834	**1.32**	**1.09**	1.02	1.16	5621	**1.43**	**1.33**	1.26	1.40
Much work—miss lunch etc., Yes (≥1 day of 2)	1579	**2.10**	**1.70**	1.58	1.84	1606	**2.24**	**2.24**	2.09	2.40
Far too much to do, Yes (agree, partly agree)	1037	**2.37**	**1.77**	1.62	1.93	1604	**2.53**	**2.37**	2.21	2.53
Job control										
Determine work pace, No ( ≤1/10 of time)	1441	**1.58**	**1.43**	1.32	1.54	3064	**1.57**	**1.41**	1.33	1.49
Determine working duties, No (no, not at all)	1126	**1.17**	1.01	0.92	1.10	2423	**1.22**	1.05	0.98	1.12
Participate in decisions, No (≤mostly not)	2188	**1.21**	**1.09**	1.01	1.17	3199	**1.11**	1.05	0.99	1.11
Influence, No (disagree, partly disagree)	2149	**1.47**	**1.38**	1.29	1.47	2613	**1.66**	**1.57**	1.48	1.67
Job support from supervisors, Mostly not—never	3278	**1.78**	**1.69**	1.59	1.80	3082	**1.77**	**1.70**	1.60	1.79
Job support from colleagues, Mostly not—never	1728	**1.63**	**1.61**	1.50	1.73	1191	**1.51**	**1.54**	1.42	1.67
Work ability ^e^										
High, 10 points	2528	1	1			3183	1	1		
9 points	634	1.02	0.96	0.85	1.08	727	**1.29**	**1.22**	1.08	1.37
8 points	1248	**1.38**	**1.17**	1.06	1.30	1436	**1.51**	**1.31**	1.19	1.44
7 points	585	**1.88**	**1.43**	1.24	1.66	652	**1.98**	**1.41**	1.22	1.62
Low, 1–6 points	332	**2.48**	**1.58**	1.28	1.94	574	**3.13**	**1.87**	1.58	2.33

^a^ All individuals in the study group (*n* = 45,098). ^b^ Number of cases with presenteeism, two times or more. ^c^ Model 1: Odds ratio (OR) of presenteeism with 95% confidence interval (CI), adjusted for age (continuous variable) and year of interview. ^d^ Model 2: Odds ratio (OR) of presenteeism with 95% confidence interval (CI) adjusted for education, employment sector, country of birth, physical work (2 variables), three health symptoms, long-term sickness absence, disability pension, age (continuous variable), and year of interview. Bold = statistically significant at the *p* < 0.05 level. ^e^ Sub-sample 2007–2013, *n* = 24,902.

**Table 4 ijerph-17-02419-t004:** Associations between presenteeism and sex, psychosocial variables, and work ability, stratified by occupation ^a^. Number of cases and odds ratios (ORs) with 95% confidence intervals (CIs) are presented.

		Presenteeism													
		All Other *n* = 37,125				Nursing Professionals *n* = 1716					Care Assistants *n* = 6257				
	*N* ^b^	OR ^c^	OR ^d^	CI	CI	*N* ^b^	OR ^c^	OR ^d^	CI	CI	N ^b^	OR ^c^	OR ^d^	CI	CI
Sex															
Man	10,085	1	**1**			72	1	1			297	1	1		
Women	8889	**1.10**	0.96	0.91	1.00	823	0.98	0.85	0.58	1.24	2591	1.11	0.87	0.73	1.05
Job Demands															
Stressful—no time to think Yes (≥3/4 of working time)	2875	**2.33**	**1.63**	1.54	1.73	202	**2.32**	**1.70**	1.34	2.14	454	**2.28**	**1.58**	1.37	1.83
Attention/concentration, Yes (≥3/4 of time)	15,885	**1.39**	**1.15**	1.09	1.20	481	**1.51**	**1.28**	1.04	1.59	1485	**1.34**	**1.13**	1.01	1.26
Much work—miss lunch etc. Yes (≥1 day of 2)	7670	**2.19**	**1.73**	1.63	1.82	182	**2.15**	**1.51**	1.19	1.93	213	**2.39**	**1.54**	1.28	1.86
Far too much to do Yes (agree, partly agree)	6643	**2.46**	**1.78**	1.67	1.89	109	**2.86**	**1.97**	1.50	2.61	300	**2.58**	**1.72**	1.46	2.03
Job Control															
Determine work pace No (≤1/10 of time)	7649	**1.62**	**1.27**	1.20	1.34	329	**1.29**	1.14	0.92	1.41	986	**1.46**	**1.21**	1.08	1.36
Determine working duties No (no, not at all)	4906	**1.21**	0.97	0.91	1.04	298	1.07	0.93	0.75	1.17	938	**1.17**	1.07	0.95	1.21
Participate in decisions, No (≤mostly not)	8800	**1.16**	0.98	0.93	1.03	264	1.16	1.07	0.85	1.34	878	1.10	1.01	0.90	1.14
Influence, No (disagree, partly disagree)	8997	**1.53**	**1.25**	1.19	1.32	211	**1.92**	**1.61**	1.28	2.02	664	**1.76**	**1.51**	1.34	1.71
Support from supervisors															
Mostly not—never	12538	**1.76**	**1.42**	1.36	1.49	257	**1.76**	**1.40**	1.12	1.75	749	**1.72**	**1.44**	1.28	1.63
Support from colleagues															
Mostly not—never	6196	**1.57**	**1.33**	1.25	1.41	51	**1.87**	1.37	0.91	2.08	201	**1.40**	1.17	0.94	1.45
Work ability ^e^															
High, 10 points	4842	1	**1**			216	1	1			653	1	1		
9 points	1174	**1.12**	1.06	0.97	1.16	48	1.03	0.93	0.59	1.45	139	**1.43**	**1.36**	1.03	1.81
8 points	2232	**1.42**	**1.23**	1.14	1.33	67	1.15	0.92	0.61	1.38	385	**1.63**	**1.44**	1.18	1.76
7 points	1010	**1.89**	**1.43**	1.28	1.60	37	**1.75**	1.12	0.63	1.98	190	**2.10**	**1.51**	1.14	2.00
Low, 1–6 points	725	**2.74**	**1.68**	1.45	1.94	28	**6.43**	**3.82**	1.57	10.77	153	**3.21**	**1.97**	1.40	2.79

^a^ All individuals in the study group (*n* = 45,098). ^b^ Number of cases with presenteeism, two times or more. ^c^ Model 1; Odds ratio (OR) of presenteeism with 95% confidence interval (CI), adjusted for age (continuous variable) and year of interview. ^d^ Model 2; Odds ratio (OR) of presenteeism with 95% confidence interval (CI), adjusted for education, sector of employment, country of birth, physical work (2 variables), three health symptoms, long-term sickness absence, disability pension, age (continuous variable), and year of interview. ^e^ Sub-sample 2007–2013, *n* = 24,902. Bold = statistically significant at the *p* < 0.05 level.

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
