# Peer review of "Presenteeism, Psychosocial Working Conditions and Work Ability among Care Workers—A Cross-Sectional Swedish Population-Based Study"

_ijerph, 2020, doi:10.3390/ijerph17072419_

Round 1
Reviewer 1 Report
This paper investigates presenteeism among Swedish nursing professionals and care assistants, focusing on how psychosocial working conditions and work ability were associated with presenteeism. Data pooled from cross-sectional bi-annual Swedish Work Environment Surveys 2001-2013, are analyzed (ca N= 1700 nurses, 6200 care assistants, 37,000 all other occupations). Presenteeism was measured by dichotomizing answers to a four-level question about number of times last 12 months the individual had gone to work although he/she should not due to medical conditions. Psychosocial working conditions were measured by questions about job demands, job control and job support, while “current work ability” was measured by a 1-10 scale. Potential confounder variables were sex, age, education etc., plus answers about physical working conditions, health-related symptoms, and receipt of disability pension the year after the survey. Multiple logistic regression, both univariate and adjusted for confounders, were used. Care assistants, but not nurses, were found to have higher levels of presenteeism than the average of all other occupations. In general, women were at higher risk. Negative assessments of psychosocial working condition, but also of physical working conditions and own health, as well as low self-reported “current work ability”, were associated with more presenteeism – more or less in the same way among all the three analyzed occupational categories.
The topic of the paper is certainly timely; currently, media report instances that corona-infected health personnel have transmitted the virus to patients. The data are good (large samples, good response rates), and all analyzed aspects (presenteeism, psychosocial working conditions, etc.) seem reasonably and adequately indicated/measured by the survey questions. The data are analyzed competently by standard multiple logistic regression models (most included variables have been dichotomized). The paper’s summary of the findings (in the Abstract and the Discussion section) corroborates well with the results in the analyses. The paper is overall well and transparently written.
Thus, since topic is relevant, data good, analyses adequate and abstract/summary in line with findings, publication may of course be legitimate and recommended.
However, since a reviewer should “speak out his/her mind”, this reviewer has also some concerns. Presenteeism, in general as well as among various types of health personnel, has been examined for years, in different contexts – as the authors themselves note (see Abstract). Many findings and interpretations have been previously presented. In view of this, the paper seems somewhat unoriginal and unambitious, perhaps “unanalytical” (meaning few efforts are made in order to understand the findings). It confirms many previous empirical results, with a large material and in a decent way (the Discussion notes that its results are similar to findings in a number of previous studies). But when facing a new study, one would like to ask: How does this study extend knowledge? Perhaps that presenteeism among nurses in this study does not deviate from the average of all other occupations? But since “all other occupations” contains very different occupations, it is unclear what this means. The association between self-reported work ability has perhaps not been investigated previously in Sweden, but – according to the Discussion – similar results have been found in many other contexts. The paper reports that mixed results as to gender differences in presenteeism exist; this study finds that the initial (univariate) higher presenteeism among women disappeared when adjusting for all other variables, but does not attempt to explain why. The policy conclusion that “efforts should be made to improve psychosocial working environments” is certainly right, but how? The study would have been more interesting if unknown or challenging questions had been addressed. What issues that should be, cannot be defined by this reviewer, of course, but I wonder, for instance: Why higher presenteeism among care assistants than among nurses? Are there time trends – gradually increasing presenteeism over time (this is perhaps impossible to analyze with these data which only spans some 10-12 years). Are the younger nurses or care assistants more prone to presenteeism than older nurses, and if so – why? Employment sector (the private/public distinction) is only used as a control variable – this may be a controversial policy/political question, but could the data be used for interesting approaches to this? This reviewer does not claim that the above-mentioned ideas are viable, but the present study, although it presents nothing wrong, does appear as somewhat “passive”, so to speak, and not oriented towards trying to solve, or illuminate, questions in the research frontier of the presenteeism complex.***
Author Response
Dear Reviewer 1. Thank you very much for the valuable comments and suggestions, which have been very helpful in our revision of the manuscript. Please find our responses below.
Comments and Suggestions for Authors
This paper investigates presenteeism among Swedish nursing professionals and care assistants, focusing on how psychosocial working conditions and work ability were associated with presenteeism. Data pooled from cross-sectional bi-annual Swedish Work Environment Surveys 2001-2013, are analyzed (ca N= 1700 nurses, 6200 care assistants, 37,000 all other occupations). Presenteeism was measured by dichotomizing answers to a four-level question about number of times last 12 months the individual had gone to work although he/she should not due to medical conditions. Psychosocial working conditions were measured by questions about job demands, job control and job support, while “current work ability” was measured by a 1-10 scale. Potential confounder variables were sex, age, education etc., plus answers about physical working conditions, health-related symptoms, and receipt of disability pension the year after the survey. Multiple logistic regression, both univariate and adjusted for confounders, were used. Care assistants, but not nurses, were found to have higher levels of presenteeism than the average of all other occupations. In general, women were at higher risk. Negative assessments of psychosocial working condition, but also of physical working conditions and own health, as well as low self-reported “current work ability”, were associated with more presenteeism – more or less in the same way among all the three analyzed occupational categories.
The topic of the paper is certainly timely; currently, media report instances that corona-infected health personnel have transmitted the virus to patients. The data are good (large samples, good response rates), and all analyzed aspects (presenteeism, psychosocial working conditions, etc.) seem reasonably and adequately indicated/measured by the survey questions. The data are analyzed competently by standard multiple logistic regression models (most included variables have been dichotomized). The paper’s summary of the findings (in the Abstract and the Discussion section) corroborates well with the results in the analyses. The paper is overall well and transparently written.
Thus, since topic is relevant, data good, analyses adequate and abstract/summary in line with findings, publication may of course be legitimate and recommended.
Response: Thank you!
However, since a reviewer should “speak out his/her mind”, this reviewer has also some concerns. Presenteeism, in general as well as among various types of health personnel, has been examined for years, in different contexts – as the authors themselves note (see Abstract). Many findings and interpretations have been previously presented. In view of this, the paper seems somewhat unoriginal and unambitious, perhaps “unanalytical” (meaning few efforts are made in order to understand the findings). It confirms many previous empirical results, with a large material and in a decent way (the Discussion notes that its results are similar to findings in a number of previous studies). But when facing a new study, one would like to ask: How does this study extend knowledge?
Response: One new finding concerns the fact that a large number of nurses and care assistants who go to work despite being ill have a low work ability. This is now emphasized in the discussion and in the conclusion.
Perhaps that presenteeism among nurses in this study does not deviate from the average of all other occupations? But since “all other occupations” contains very different occupations, it is unclear what this means.
Response: It is true that the “all other occupations” group is very heterogeneous. The fact that nurses do not deviate from this group, as the care assistants do, implicate that there are differences even within the health and social care sector. We have now addressed this issue in the discussion.
The association between self-reported work ability has perhaps not been investigated previously in Sweden, but – according to the Discussion – similar results have been found in many other contexts.
Response: Previous studies have focused on other occupations and other indicators of work ability. This has now been clarified in the discussion.
The paper reports that mixed results as to gender differences in presenteeism exist; this study finds that the initial (univariate) higher presenteeism among women disappeared when adjusting for all other variables, but does not attempt to explain why.
Response: Our interpretation of the fact that the gender differences, which were only some few percentages, disappeared when other factors were adjusted for is that the confounders have different associations with presenteeism among women and men. As our focus was on working conditions this may be seen to indicate that female and male health care employees (as well as other employees) have different working conditions, but also different educational level.
The policy conclusion that “efforts should be made to improve psychosocial working environments” is certainly right, but how?
Response: Thank you for pointing this out. We have now given some more specific suggestions for policies.
The study would have been more interesting if unknown or challenging questions had been addressed. What issues that should be, cannot be defined by this reviewer, of course, but I wonder, for instance: Why higher presenteeism among care assistants than among nurses? Are there time trends – gradually increasing presenteeism over time (this is perhaps impossible to analyze with these data which only spans some 10-12 years).
Response: Our interpretation of the higher presenteeism among care assistants than among nurses and the all other occupational group is that care assistants have more problematic working conditions. National information on sickness presenteeism is available in Sweden only from 2001, but generally there have been very small changes over time. We have now clarified this in the discussion and included at reference to Statistics Sweden, Swedish Work Environment Survey, to substantiate the argument.
Are the younger nurses or care assistants more prone to presenteeism than older nurses, and if so – why? Employment sector (the private/public distinction) is only used as a control variable – this may be a controversial policy/political question, but could the data be used for interesting approaches to this?
Response: No, the opposite is true, older nurses and care assistants as well as “all other occupations” are more prone presenteeism. This has also been found in other studies. We have now added a comment in the discussion section about this, where we speculate that this may have to do with differences in health and employment security in the different age groups. Unfortunately there were too few employed in the private sector to conduct proper stratified analyses.
This reviewer does not claim that the above-mentioned ideas are viable, but the present study, although it presents nothing wrong, does appear as somewhat “passive”, so to speak, and not oriented towards trying to solve, or illuminate, questions in the research frontier of the presenteeism complex.

Reviewer 2 Report
Thank you for the opportunity to review the revised manuscript “Presenteeism, psychosocial working conditions, and work ability among care workers - A cross-sectional Swedish population-based study” for International Journal of Environmental Research and Public Health. This is a thought-provoking manuscript, and I enjoyed reviewing it. The data are unique and allow the author(s) to examine (in an exploratory fashion) how psychosocial working conditions and perceived work ability are associated presenteeism among nursing professionals/ care assistants in Sweden. Overall, it was clearly written and thought provoking. I do, however, have some comments on how the manuscript could be improved.
- I’d like to know more about the origins of the data used in the paper. Specifically, the authors need to clarify if these data are repeated cross-sectional trend data or are these panel data. This is an important point to clarify. In my view the scholarship of this work would be strengthened considerably if the author(s) would be more explicit explaining the nature of their data. Without doing so, I am unable to accurately interpret the results.
- Likewise, this report would be strengthened by clarifying both -1- the response options for the individual survey items, and -2- the author’s coding decisions.
- Small concern: the font size fluctuates throughout the paper.
In all, this paper has the potential to make a nice contribution to the literature, and I look forward to seeing a revised version of in.
Author Response
Dear Reviewer 2. Thank you very much for the valuable comments and suggestions, which have been very helpful in our revision of the manuscript. Please find our responses below
Comments and Suggestions for Authors
Thank you for the opportunity to review the revised manuscript “Presenteeism, psychosocial working conditions, and work ability among care workers - A cross-sectional Swedish population-based study” for International Journal of Environmental Research and Public Health. This is a thought-provoking manuscript, and I enjoyed reviewing it. The data are unique and allow the author(s) to examine (in an exploratory fashion) how psychosocial working conditions and perceived work ability are associated presenteeism among nursing professionals/ care assistants in Sweden. Overall, it was clearly written and thought provoking. I do, however, have some comments on how the manuscript could be improved.
- I’d like to know more about the origins of the data used in the paper. Specifically, the authors need to clarify if these data are repeated cross-sectional trend data or are these panel data. This is an important point to clarify. In my view the scholarship of this work would be strengthened considerably if the author(s) would be more explicit explaining the nature of their data. Without doing so, I am unable to accurately interpret the results.
Response: Thank you for this comment. The surveys were based on year-specific random sample, not panel data or repeated measurement. We have clarified this in the method-section.
- Likewise, this report would be strengthened by clarifying both -1- the response options for the individual survey items, and -2- the author’s coding decisions.
Response: We have clarified this. A cross-sectional study on seven cohorts of the Swedish employed population collected from surveys between 2001 and 2013 was conducted.
We follow the response scales from the original SWES. Designed in 1989 and conducted every second year since then. We have chosen to dichotomize according to a commonly used conventional method with the upper quartile as the benchmark for negative load. This is also to reach consistency with several other articles in the project. We also refer to two previous studies within the project, that have more detailed information about the response alternatives and the coding procedure [25, 26].
- Small concern: the font size fluctuates throughout the paper.
Response: Thank you for pointing this out. We have now corrected this.
In all, this paper has the potential to make a nice contribution to the literature, and I look forward to seeing a revised version of in.

Reviewer 3 Report
Dear authors,
The study topic is interesting and the manuscript is easy to read. I have some recommendations about it:
- The abstract should indicate the number of healthcare workers included in the analysis.
- The study design must be indicated at the beginning of section 2.1
- The size of the letter is different in some parts of the text
- Line 131-132. There is a mistake it sais that it was "coded into five categories" but it has only 2 categories.
- Does Table 2 include information about healthcare workes also? It is not clear.
Author Response
Dear Reviewer 3. Thank you very much for the valuable comments and suggestions, which have been very helpful in our revision of the manuscript. Please find our responses below.
Comments and Suggestions for Authors
Dear authors,
The study topic is interesting and the manuscript is easy to read. I have some recommendations about it:
- The abstract should indicate the number of healthcare workers included in the analysis.
Response: Good suggestion. We have included number of healthcare workers (nurses, care assistants, but also the all other occupation group) in the abstract.
- The study design must be indicated at the beginning of section 2.1
Response: We agree and have now included the concept cross-sectional design in section 2.1
- The size of the letter is different in some parts of the text
Response: Thank you for pointing this out. We have now corrected this.
- Line 131-132. There is a mistake it says that it was "coded into five categories" but it has only 2 categories.
Response: We apologize for not being clear on this. Responses were originally given on a ten-point scale and coded by us into five categories, “1-6 points” (low work ability), “7”, “8”, “9”, remained in the original code and “10 points” represented high work ability. We have now clarified this in more detail.
- Does Table 2 include information about healthcare workers also? It is not clear.
Response: Yes, information about healthcare workers are included in the table 2. We have changed to individuals instead of cases in the table title, such as in the footnote.

Round 2
Reviewer 1 Report
In the revised manuscript, the authors have broadened their discussion on possible interpretations of their results, which has improved the paper. Overall, to this reviewer, the paper seems now an interesting contribution to the research field, and I have no further comments.
However, as to proof reading: perhaps look into the sentences below:
Line 63: “…cross-sectional study based on seven surveys working population in Sweden between…”
Lines 169-172: “… included in the SWES (spanning 2001-2013) was operationalized as the number of spells of the individuals medically certified sickness absence lasting 60 days or more, as recorded in the Swedish Social Insurance registers and obtained from the LISA database. The categories are ‘No’ (<60 days) and ‘Yes’ (>60 days). …”
Line 294: “… decades in Sweden [26]. This can indicate that there may be also be compensating factors at work …”
***